# The Use of Antiviral Agents against SARS-CoV-2: Ineffective or Time and Age Dependent Result? A Retrospective, Observational Study among COVID-19 Older Adults [note 1]

**DOI:** 10.3390/jcm10040686

**Published:** 2021-02-10

**Authors:** Antonio Desai, Giuseppe Caltagirone, Sharon Sari, Daria Pocaterra, Maria Kogan, Elena Azzolini, Victor Savevski, Filippo Martinelli-Boneschi, Antonio Voza

**Affiliations:** 1Emergency Department, Humanitas Clinical and Research Center, IRCCS, 20089 Milan, Italy; giuseppe.caltagirone@humanitas.it (G.C.); maria.kogan@humanitas.it (M.K.); antonio.voza@humanitas.it (A.V.); 2Department of Biomedical Sciences, Humanitas University, 20090 Pieve Emanuele, Italy; elena.azzolini@humanitas.it; 3Internal Medicine Department, Geriatrics, Santa Margherita Rehabilitation and Cure Institute, ASP, 27100 Pavia, Italy; shari828@gmail.com; 4Department of Infectious Diseases, Humanitas Clinical and Research Center, IRCCS, 20089 Milan, Italy; daria.pocaterra@humanitas.it; 5Health Directorate, Humanitas Clinical and Research Center, IRCCS, 20089 Milan, Italy; 6Artificial Intelligence Center, Humanitas Clinical and Research Center, IRCCS, 20089 Milan, Italy; victor.savevski@humanitas.it; 7Dino Ferrari Centre, Neuroscience Section, Department of Pathophysiology and Transplantation (DEPT), University of Milan, 20122 Milan, Italy; filippo.martinelli@unimi.it; 8Neurology Unit and MS Centre, Fondazione IRCCS Ca’ Granda Ospedale Maggiore, Policlinico, 20122 Milan, Italy

**Keywords:** age, antivirals, COVID-19, darunavir/cobicistat, geriatric, gerontology, hydroxychloroquine, lopinavir/ritonavir, SARS-CoV-2, timing

## Abstract

Background: Our aim was to investigate the impact of therapeutics with antiviral activity against severe acute respiratory syndrome coronavirus 2 (SARS-CoV-2) on mortality of older adults affected by coronavirus disease 2019 (COVID-19), taking into consideration the time interval from symptoms onset to drugs administration. Methods: Data from 143 COVID-19 patients over 65 years of age admitted to the Humanitas Clinical and Research Center Emergency Department (Milan, Italy) and treated with Lopinavir/ritonavir (LPV/r) or Darunavir/cobicistat (DVR/c) associated to Hydroxychloroquine (HCQ) were retrospectively analyzed. Statistical analysis was performed by using a logistic regression model and survival analysis to assess the role of different predictors of in-hospital mortality, including an early (<6 days from symptoms onset) vs. late treatment onset, signs and symptoms at COVID-19 presentation, type of antiviral treatment (LPV/r or DVR/c) and patients’ age (65–80 vs. >80 years old). Results: Multivariate analysis showed that an older age (OR: 2.54) and dyspnea as presenting symptom (OR: 2.01) were associated with higher mortality rate, whereas cough as presenting symptom (OR: 0.53) and a timely drug administration (OR: 0.44) were associated with lower mortality. Survival analysis demonstrated that the timing of drug administration had an impact on mortality in 65–80 years-old patients (*p* = 0.02), whereas no difference was seen in those >80 years-old. This impact was more evident in patients with dyspnea as primary symptom of COVID-19, in whom mortality decreased from 57.1% to 38.3% due to timely drug administration (OR: 0.5; *p* = 0.04). Conclusions: There was a significant association between the use of a combined antiviral regimen and HCQ and lower mortality, when timely-administered, in COVID-19 patients aged 65–80 years. Our findings support timely treatment onset as a key component in the treatment of COVID-19.

## 1. Introduction

During the severe acute respiratory syndrome coronavirus 2 (SARS-CoV-2) pandemic, the causal agent of coronavirus disease 2019 (COVID-19), several drugs were investigated in randomized trials worldwide including antiviral agents and antimalarial medications [1]. However, as yet, the World Health Organization (WHO) claimed that there is no specific treatment recommended to prevent or treat COVID-19, and isolation, quarantine, and infection-control measures remain the gold standard to reduce the spread of the virus.

A Human Immunodeficiency Virus (HIV) protease inhibitor, Lopinavir/Ritonavir (LPV/r), has been a suitable candidate for SARS-CoV-2 infection treatment given its in vitro efficacy against the severe acute respiratory syndrome (SARS) associated coronavirus [1,2]. A similar consideration was also applied to Darunavir/Cobicistat (DVR/c) [2,3,4,5]. Hydroxychloroquine (HCQ), on the other hand, has been used for decades to treat malaria and other autoimmune diseases. Its anti-inflammatory and immunomodulating properties have made HCQ a feasible therapeutic strategy to consider for the treatment of COVID-19, alone or in association with other medications [6,7,8].

Notwithstanding, pivotal studies have failed to prove these drugs to be effective against COVID-19, and no prior study has considered the timing of drug administration from symptom onset in older adults [9,10,11,12,13]. 

In the light of the fact that the risk for severe illness due to COVID-19 increases with age and SARS-CoV-2 infection-related deaths were mostly seen in older adults (80% of cases) [14], we performed a retrospective observational study in older adults (considered as adults above 65 years of age), to investigate whether a combination of therapeutics with antiviral activity against SARS-CoV-2 could be truly considered ineffective in the treatment of SARS-CoV-2 infection. We decided to focus our analysis on these drugs, despite the fact they have been previously reported to be ineffective, to evaluate whether timely drug administration can have an impact on the natural course of the disease.

Therefore, the main objective of the study was to investigate whether a timely administration of a combined regimen of HCQ and LPV/r or DVR/c, related to the symptom onset, could improve patients’ outcome in terms of mortality. A secondary endpoint was to estimate the impact of a timely drug administration in terms of disease progression and need to Intensive Care Unit (ICU) admission in hospitalized patients. Association between symptom type at disease onset, whether cough, dyspnea, fever, gastrointestinal (GI) symptoms, and/or anosmia and ageusia, and mortality were also tested as secondary endpoints.

## 2. Materials and Methods

### 2.1. Study Design, Participants, and Data Collection

This single-center observational study was conducted at the Emergency Department (ED) of the Humanitas Clinical and Research Center, Milan, Italy (HUMANITAS). The study was approved by the ethical committee of HUMANITAS and performed in accordance with the principles of the Helsinki declaration (Protocol nr. 369/20; approved on: 22 April 2020) [15]. 

We retrieved and analyzed data of 143 older adults (patients over 65 years of age) admitted to our ED from February 21st, 2020 to April 14th, 2020. All the patients included in our study presented with symptoms attributable to SARS-CoV-2 infection on hospital admission and were diagnosed with COVID-19 according to WHO interim guidance [16]. Real Time Polymerase Chain Reaction (RT-PCR) was performed on a nasopharyngeal swab in all subjects to establish a laboratory confirmation for SARS-CoV-2 infection. Additional inclusion criteria were to receive a combined treatment with HCQ and LPV/r or DVR/c according to the Italian Health Ministry rules and local protocols at HUMANITAS and access to the full set of clinical and radiological data. The ED discharge or the need for hospitalization was assessed on a case-by-case approach.

Clinical electronic medical records were reviewed for all patients who tested positive for SARS-CoV-2 infection. Data were collected and stored in a dedicated database built solely for this purpose. Any missing or uncertain records were clarified through direct communication with the relevant health-care providers and family members.

The following variables were collected: age, gender, comorbidities, chronic treatment, the time interval between symptoms onset and drug administration, type of symptom at the presentation of COVID-19 (fever, cough, dyspnea, gastrointestinal disturbances (GI), hyposmia/hypogeusia), treatment provided during Emergency Department (ED) stay (antiviral agents, antibacterial agents, Hydroxychloroquine, antibiotics, and low molecular weight heparin (LWMH) therapy), days of hospitalization, pulmonary computed tomography (CT) scan categorized into the appearance of ground-glass opacity, consolidation, lymphadenopathy, and pleural effusion, and clinical outcome of patients including in-hospital mortality and ICU transfer as reported in a previous paper [17]. Data were made available only to authorized personnel, stored on a local server, and retrieved for this analysis.

### 2.2. Lopinavir/Ritonavir (LPV/r), Darunavir/Cobicistat (DVR/c), Hydroxychloroquine (HCQ), and Need for Hospitalization

Antiviral therapy was administered as follows: Lopinavir/Ritonavir 400 mg/100 mg orally twice daily or Darunavir/Cobicistat 800/150 mg orally once daily, for a period of five to twenty days. The duration of treatment was based on the clinical response. Drug selection was based solely on availability. HCQ 200 mg orally twice a day was started in all patients with confirmed SARS-CoV-2 infection. Treatment duration was between 5 and 20 days, depending on the patients’ clinical response. Patients with prolonged QT intervals on the electrocardiogram were excluded due to the potential proarrhythmic effect of the drug.

The need for hospitalization was evaluated case-by-case, considering multiple factors such as age, clinical status, respiratory parameters, and comorbidities. CURB65, which is a clinical algorithm validated for predicting mortality in CAP (community-acquired pneumonia) and infection of any site, and pneumonia severity index (PSI) scores was also used but not included in these analyses [18,19].

### 2.3. Statistical Analysis

Categorical variables were represented in terms of frequency distributions, while quantitative variables were described as mean and standard deviation (SD) or median and interquartile range (IQR). 

The primary outcome of the study is to investigate whether a timely administration of a combined regimen of HCQ and LPV/r or DVR/c, related to the symptom onset, could improve patients’ outcome in terms of mortality. Assuming mortality of 43.5% detected in a previous study on 3191 COVID-19 patients >65 years (prevalence ranges from 33.7% in those between 65 and 80 years to 53.5% in those >80 years) [20], the required sample size to achieve an 80% power (*β* = 0.2) at *α* = 0.05 to detect a difference in mortality between early vs. late drug start with 1 to 1 allocation is 154 patients assuming an OR of 2.5, and 110 patients assuming an OR of 3.0 [21]. We performed univariate analyses using the chi-square test for independence for categorical variables and unpaired *t*-test (or, if assumptions were not met, Mann–Whitney’s test) for quantitative variables. Outcomes of interest were mortality and transfer to ICU expressed as categorical variables or as survival time. 

Survival analysis was performed using the Kaplan–Meier approach in the entire cohort of patients, as well as in patients between 65 and 80 years and in those aged >80 years. The proportionality of hazard was tested by means of the log–rank test. A multivariate analysis using a logistic regression model was performed with mortality as a dependent variable and as predictors, those variables which were significant with *p* < 0.1 at univariate testing with a backward selection based on likelihood ratio to find the most parsimonious model. Tested variables as predictors were: age, gender, type of antivirals used, LMWH use, type of symptom at presentation (fever, cough, dyspnea, GI problems, hyposmia/hypogeusia), and the time interval between symptoms onset and drug administration.

In inferential testing, *p* < 0.05 was considered statistically significant. Analyses were performed with IBM SPSS software version 26.0.

## 3. Results

We focused our analyses on 143 older adults treated with LPV/r and HCQ or DVR/c and HCQ, representing 24.8% of the total sample of COVID-19 patients who accessed the ED and 47.9% of those >65 years, for whom we retrieved the date of symptom onset which was within 6 days in 50.3% and later in 49.7%.

The mean age was 76.2 years, 68.5% of patients were between 65 and 80 years, and 31.5% above 80 years of age. The male to female ratio was 2.5 to 1. In terms of the outcome, in-hospital mortality was 42%, and 7% of patients were transferred to ICU. 

When we stratified patients into two groups (65–80 vs. >80 years), there was a significant difference in mortality (33.7% vs. 60%; *p* = 0.003) and in the percentage of patients who were transferred to the ICU (10.2% vs. 0%; *p* = 0.03). 

Presenting symptoms of COVID-19 were fever in 88.1% of patients, dyspnea in 67.1%, cough in 53.8%, whereas GI disturbances and hyposmia/hypogeusia were seen in 18.9% and 5.6% of patients, respectively (Table 1). In 81.8% of cases, clinical onset was characterized by more than one symptom. In terms of CT scan, all patients had a ground-glass appearance, 33.6% a consolidation, while lymphadenopathy was seen in 35.5%, and pleural effusion in 14.7% of cases (Table 1).

When we stratified patients into the two age groups, we noted a significantly higher prevalence of cough as presenting symptom in 65–80 years old group compared to >80 years (62.2% vs. 35.6%; *p* = 0.003) and a lower incidence of dyspnea complaint (62.2% vs. 77.8%; *p* = 0.06) (Table 1). No difference was found in CT findings between the two age groups.

Moreover, no difference was found between the two age groups in terms of treatment onset, which was within 6 days from symptom onset in 49% of patients in patients aged 65 to 80 years and 53.3% in patients aged over 80 years old. Moreover, no differences were found in patients treated with LPV/r antiviral scheme associated with HCQ compared to DVR/c associated with HCQ (69.4% vs. 64.4%). A significant difference was found in terms of LMWH use between the two age groups (82.7% of pts. between 65 to 80 versus 100% in patients over 80 years old (*p* = 0.003)) (Table 1). 

When assessing the role of different variables on mortality, we found that age category, cough and dyspnea as presenting symptoms, and time to drug start were associated with mortality at univariate analysis. Type of antiviral treatment had no impact on mortality (40.2% for LPV/r vs. 45.7% for DVR/c; *p* = 0.54) (Table 2). 

When tested at multivariate analysis, the final most parsimonious model included an older age (OR: 2.54; 95% CI: 1.2–5.6; *p* = 0.03) and dyspnea as presenting symptom (OR: 2.01; 95% CI: 0.9–4.4; *p* = 0.08) as risk factors, whereas cough as presenting symptom (OR: 0.53; 95% CI: 0.2–1.1; *p* = 0.09) and a timely treatment administration (OR: 0.44; 95% CI: 0.2–0.9; *p* = 0.02) were protective factors (Table 2). 

When we stratified patients according to their age category, there was an effect of early treatment administration on mortality in those aged 65 to 80 years-old with a median survival of 44 vs. 20 days (*p* = 0.02; log–rank test), while no effect was seen in patients over 80 years old (median survival: 11 vs. 12 days; *p* = 0.66; log–rank test) (Figure 1 and Figure 2). Figure 3 depicts the impact of an early versus late drug administration in patients who experience dyspnea with a mortality ranging from 38.1% to 57.1% (OR: 0.4; 95% CI: 0.17–0.99; *p* = 0.04), whereas in patients with no dyspnea at disease onset the difference in mortality due to an early versus late treatment onset was less relevant (24% vs. 36.4%; OR: 0.5; 95% CI: 0.13–1.81; *p* = 0.03). 

## 4. Discussion

This single-center, observational study has shown that an early (within 6 days of symptoms onset) administration of therapeutics with antiviral activity against SARS-CoV-2 associated with Hydroxychloroquine could be effective in the treatment of COVID-19 in patients from 65 to 80 years of age, whereas it has been found to be less effective in patients over 80 years old. Noteworthy, the time interval from symptom presentation to treatment onset has been identified as a critical component, which we assume contributed to a more favorable outcome [19]. These results could be explained by the fact that the first stage of SARS-CoV-2 infection (also known as Stage I), in which the virus replication is taking place, is more sensitive to antiviral agents, and either HCQ and LPV/r or DVR/c can potentially decrease the viral load changing the natural course of disease [22]. As a matter of fact, the first phase is generally characterized by fever and dry cough occurrence. These were more commonly depicted as the presenting symptoms in individuals between 65 and 80 years of age and were associated with lower mortality in the current study. On the contrary, drugs administered in a later phase, such as phase two (namely the pulmonary phase in which dyspnea was the cardinal symptom) or phase 3 (also called the inflammatory phase), seemed to be less effective [17,22,23,24,25,26,27,28].

Of note, Cao reported the first open-label randomized controlled clinical trial of LPV/r therapy during the SARS-CoV-2 pandemic. Ninety-nine patients were assigned to the LPV/r group, whereas the other half was assigned to the standard-care group. The study concluded that no benefits were observed with LPV/r treatment when compared to standard care. Interestingly enough, they found that early initiation of the treatment (<12 days from symptom onset) lead to a lower mortality rate as compared to standards of care with no statistical significance [29]. An additional paper comparing LPV/r monotherapy and combined therapy with Interferon beta-1b and ribavirin showed a significant difference on virologic outcomes and clinical recovery, which was most pronounced in those who received the drug within 7 days from symptom onset [29]. Similar data were shown in another study, showing a shorter duration of viral shedding in pts. treated with LPV/r and Interferon within 5 days of symptom onset [30]. 

A more recent, multicentric (over 30 countries) mortality trial in 11,266 adults hospitalized for COVID-19 of four re-purposed antiviral drugs (Remdesivir, Hydroxychloroquine, Lopinavir (fixed-dose combination with Ritonavir) and Interferon-β1a did not find any significant results [30]. A possible explanation of the low efficacy of the abovementioned studies could be that patients were enrolled in advanced stages of the disease (e.g., stage II or III) rather than in an initial phase usually described as > 10 days after symptom onset, thus lacking the possibility to impede viral replication at an early disease stage. However, in the study, there was no evaluation of the impact of time of drug administration and no evaluation of a combined treatment of Hydroxychloroquine and an antiviral treatment as in this case. 

As of DVR/c, there are less data on its efficacy. A small trial including 30 patients with mild form was randomized to receive DRV/c for 5 days and Interferon-alpha 2b inhalation or Interferon-alpha 2b inhalation alone, with no difference in efficacy [30]. Another study in a very small sample of critically ill patients with COVID-19 showed that the Darunavir–Cobicistat therapy was found to be associated with a significant survival benefit in critically ill patients with SARS-CoV-2 infection [30].

We are not aware of any study testing the efficacy of an early administration of HCQ and anti-retroviral agents, even if out of a controlled clinical trial setting. On the other hand, we have to claim that we cannot infer from this study whether the efficacy on disease outcome was related to a combination of either one of the two antiviral drugs with HCQ or DVR/c and LPV/r alone.

Additional predictors of mortality were also investigated with multivariate analysis in this study, and age was confirmed as a key player in COVID-19 patients’ outcome. Several explanations can be provided to clarify this finding: (1) older adults are more likely to have multiple comorbidities which represent significant risk factors for COVID-19 complications [31]; (2) frailty, defined as a biological syndrome featured by cumulative declines in different physiological systems and resulting in a loss of reserves and resistance to external stressors, is more common in patients aged over 80 years; (3) immune system efficiency is known to decrease during aging, representing a physiological mechanism leading to a higher susceptibility to infections; 4) reduced mobility and autonomy can lead to a delay in access to hospital care [32,33,34,35,36,37].

Interestingly, we observed distinctive patterns when analyzing the results in terms of the presenting symptoms of SARS-CoV-2 infection. In univariate and multivariate analyses, cough was found to be associated with lower mortality, whereas dyspnea as presenting symptom with higher mortality; this could be due to the fact that its presence implicates the onset of respiratory failure typical of stage II of the disease, while cough could be present in phase I. These same symptoms occurred differently in different age categories, cough being more commonly the presenting symptom in younger patients, whereas dyspnea in older individuals. These results can be interpreted by the fact that an earlier occurrence of defense mechanisms, such as cough, by younger individuals could anticipate the awareness of the disease and the timing of drug administration. In this regard, it is worthwhile mentioning that all patients recruited in this study had to have a ground-glass appearance on the pulmonary CT scan to confirm a pulmonary involvement.

Study limitations of this observational study include the retrospective collection of data, the limited sample size, and the non-randomization scheme. The consideration that presenting symptoms at ED admission were retrieved from medical charts could explain the low prevalence of symptoms, such as anosmia and ageusia, which were perhaps not sufficiently scrutinized at the beginning of the pandemic [23,38,39,40].

In these unpredictable times of facing a rapidly spreading viral pandemic, we consider of great importance to focus on treatments that might change the natural course of the disease, in particular in high-risk patients. An important risk factor being the patient’s age, which we chose to focus on in the current study. We also consider of equal importance the urgent need to identify the molecular mechanisms underlying infection spread in specific patient subsets. Of note, it has been reported that the progression of the disease follows some pathways of diffusion and immune-escape which are often adopted by tumors, especially affecting the nervous system [41,42,43,44,45,46,47]. 

The main findings we can deduct from the current study regarding patients over 65 years of age are that symptom presentation, age category, and timely drug administration can modify the outcome in terms of mortality risk. An accurate clinical monitoring of symptoms, in particular in older adults, and a timely administration of treatment represent fundamental elements in the management of the COVID-19 pandemic in older patients. 

Statistical power calculation showed that our sample of 143 patients allowed the detection of an OR of difference in mortality between early and late drug start between 2.5 and 3.0. We are very keen to see if larger observational studies and randomized controlled clinical trials will confirm whether a timely administration of therapeutics with antiviral activity against SARS-CoV-2, which were accompanied by LWMH and antibiotics in selected patients, is effective in reducing mortality. Moreover, the use of each drug individually, rather than in combination, should be evaluated to understand their efficacy. 

However, it is important to remember that treating a disease is not the same as stopping the contagion. Compliance with current regulations is the only key to reverse the trend and avoid the spread of the virus.

## 5. Conclusions

Even if not specifically classified as a time-dependent disease, COVID-19, as many other viral conditions, seem to benefit from early intervention, particularly in patients between 65 and 80 years old. Attention to particular symptoms at patient presentation could encourage earlier intervention and drug administration. Of note, symptom onset should be carefully considered when facing COVID-19 as it can lead to an appropriate therapeutic approach.

## Figures and Tables

**Figure 1 jcm-10-00686-f001:**
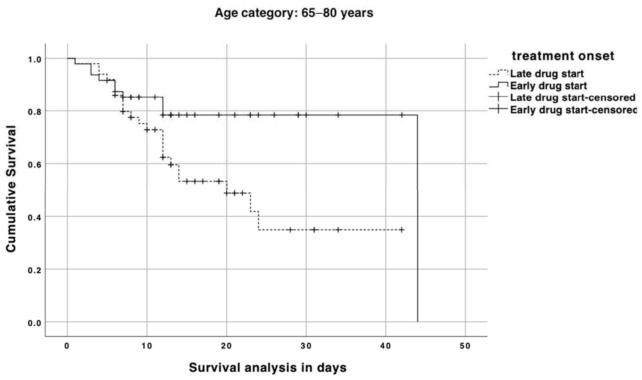
Survival analysis in coronavirus disease 2019 (COVID-19) patients with 65–80 years comparing those who started the drug within 6 days (straight line) or later (dashed line) from symptom onset. A vertical symbol on lines represents censored cases.

**Figure 2 jcm-10-00686-f002:**
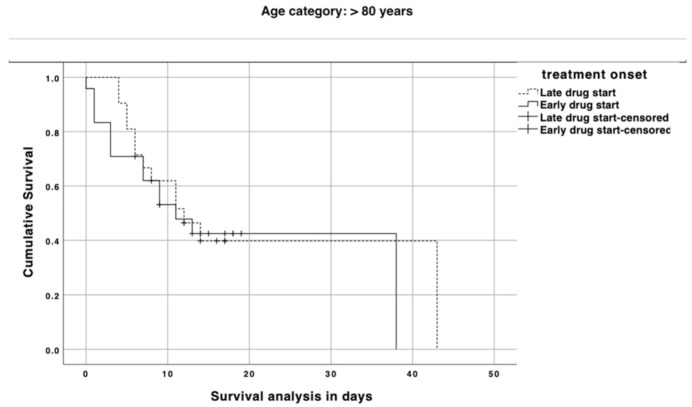
Survival analysis in COVID-19 patients with >80 years comparing those who started the drug within 6 days (straight line) or later (dashed line) from symptom onset. A vertical symbol on lines represents censored cases.

**Figure 3 jcm-10-00686-f003:**
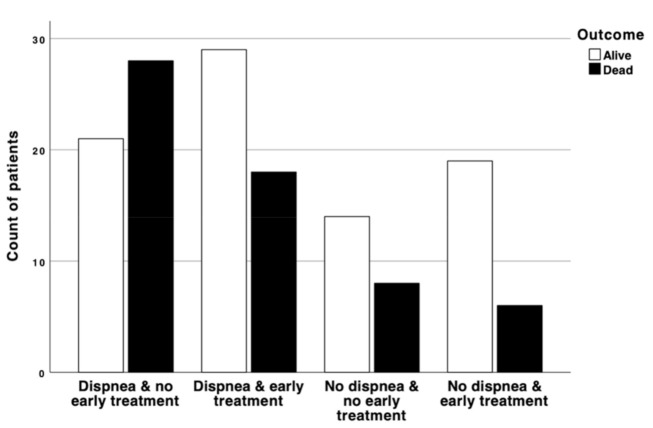
Comparison of frequency of alive (white column) and dead (black column) patients in four categories: dyspnea with late treatment; dyspnea with early treatment; no dyspnea and late treatment, and no dyspnea and early treatment.

**Table 1 jcm-10-00686-t001:** Baseline clinical features of patients stratified by age category (65–80 years and >80 years). The *p* value refers to the comparison between >80 years to 65–80 years.

Variables		Age Categories		
	Total (*n* = 143)	65–80 Years (*n* = 98)	>80 Years (*n* = 45)	*p*-Value
Age, mean (SD), min-max	76.2 (7.4) 66–93	71.9 (3.8) 66–79	85.5 (4.2) 80–93	*p* < 0.0001
Male, *n* (%)	102 (71.3%)	72 (73.5%)	30 (66.7%)	*p* = 0.40
Antivirals, LPV/r:DVR/c (% of LPV/r)	97:46 (67.8%)	68:30 (69.4%)	29:16 (64.4%)	*p* = 0.55
LMWH, *n* (%)	126 (88.1%)	81 (82.7%)	45 (100%)	***p* = 0.003**
Fever, *n* (%)	126 (88.1%)	88 (89.8%)	38 (84.4%)	*p* = 0.36
Cough, *n* (%)	77 (53.8%)	61 (62.2%)	16 (35.6%)	***p* = 0.003**
Dyspnea, *n* (%)	96 (67.1%)	61 (62.2%)	35 (77.8%)	*p* = 0.06
Gastrointestinal problems, *n* (%)	27 (18.9%)	20 (20.4%)	7 (15.6%)	*p* = 0.49
Hyposmia/hypogeusia, *n* (%)	8 (5.6%)	5 (5.1%)	3 (6.7%)	*p* = 0.71
>1 symptoms, *n* (%)	117 (81.8%)	83 (84.7%)	34 (75.6%)	*p* = 0.19
Time to drug start (<6 days), *n* (%)	72 (50.3%)	48 (49%)	24 (53.3%)	*p* = 0.63
In-hospital death, *n* (%)	60 (42%)	33 (33.7%)	27 (60%)	*p* = 0.003
ICU transfer, *n* (%)	10 (7%)	10 (10.2%)	0 (0%)	***p* = 0.03**

LPV/r: Lopinavir/Ritonavir; DVR/c: Darunavir/Cobicistat; ICU: Intensive care Unit; LMWH: Low Molecular Weight Heparin; SD: Standard Deviation. In bold significant findings.

**Table 2 jcm-10-00686-t002:** Significant predictors of in-hospital mortality at univariate and multivariate analysis. ORs and 95% CI refers to the risk of mortality according to univariate and multivariate models.

Predictors of Mortality				
	Univariate (*n* = 143)		Multivariate (*n* = 143)	
Predictors	OR (95% CI)	*p*-Value	OR (95% CI)	*p*-Value
Age: >80 vs. 65–80 years	2.95 (1.4–6.1)	0.003	2.54 (1.2–5.6)	0.03
Cough, *n* (%)	0.48 (0.24–0.94)	0.03	0.53 (0.2–1.1)	0.09
Dyspnea, *n* (%)	2.17 (1.03–4.55)	0.04	2.01 (0.9–4.4)	0.08
Time to drug start (< 6 days), *n* (%)	0.49 (0.2–0.9)	0.03	0.44 (0.2–0.9)	0.02

## Data Availability

The data presented in this study are available on request from the corresponding author. The data are not publicly available due to the limitations of the study protocol.

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
