# Peer review of "The Use of Antiviral Agents against SARS-CoV-2: Ineffective or Time and Age Dependent Result? A Retrospective, Observational Study among COVID-19 Older Adults †"

_jcm, 2021, doi:10.3390/jcm10040686_

Round 1

Reviewer 1 Report

I think this is an interesting piece of work.

My main criticism is that is not well structured. The authors seem to have identified a large numbers of interesting results, and constructed a narrative around them, to present them all. One expects studies to raise 1 main question and a number of secondary questions very clearly, and then reports results answering each one of them, going straight to the point. 

1- Abstract

Please mention the specific drugs you are working with fr the beginning and throughout the manuscript, instead of referring to the as antivirals

Line 32 Data FROM, not data ON

You can not conclude that these drugs were effective, only that there was a significant association between their use and lower mortality. Please rephrase.

Please present adjusted OR for those with dyspnoea treated with the drugs v not treated, in the abstract, not only percentages of deaths

Results in the abstract are in general not very clear. I would present ASSOCIATIONs between the different predictors and mortality, one by one, age, gender, drugs, time of drug administration, age category... with ORs of the most relevant ones.

2- Itroduction

First sentence is too long and difficult to understand

I don't understand the last lie of the intro. You look at the association between drugs and development of syptoms? Please clarify

3- Material and methods

Please avoid the ream retrospective. Clinicians collect the data prospectively. Refer to you study as observational.

4- Results

Why do you take data only of 143 patients? I don't understand. Please clarify.

These 143 patients were all admitted in the hospital wards? It is unclear to me what happened with those who were discharged home directly from the ED. Are they included in the study?

I can't see in table 1 the proportions of those who had HCQ, alone or in combination, in each age category.

You don't mention the HCQ at all in your results. Was it not one of your drugs of interest?

5- Discussion

Avoid the term retrospective.

You have to mention that you have observed outcomes on patients receiving a combination of drugs Maybe the effects you observe come only from one of them.

Please include the large systematic reviews of observational studies and clinical trials that are already available on this topic, in your bibliography. You cite very few studies, selected for unclear reasons.

Author Response

I think this is an interesting piece of work.

My main criticism is that is not well structured. The authors seem to have identified a large numbers of interesting results, and constructed a narrative around them, to present them all. One expects studies to raise 1 main question and a number of secondary questions very clearly, and then reports results answering each one of them, going straight to the point. 

1- Abstract

Please mention the specific drugs you are working with fr the beginning and throughout the manuscript, instead of referring to the as antivirals

Given the reduced sample size in our study, we included the use of anti-viral agents as a unique variable given the absence of significant difference on mortality induced by the two drugs (table 1, p=0.55). We also specified in the manuscript that we used 2 different anti-viral drugs (Lopinavir/Ritonavir (LPV/r) or Darunavir/Cobicistat (DVR/c), and, according to your comments, we better specified it in the abstract.

Line 32 Data FROM, not data ON

We made the modification in the text

You can not conclude that these drugs were effective, only that there was a significant association between their use and lower mortality. Please rephrase.

We rephrase the sentence in the abstract: there was a significant association between the use of a combined regimen with Hydroxychloroquine and Lopinavir/ritonavir or Darunavir/cobicistat and lower mortality.

Please present adjusted OR for those with dyspnoea treated with the drugs v not treated, in the abstract, not only percentages of deaths

As requested, we added the adjusted OR in the abstract (OR: 0.4). The 95% CI were 0.17-0.99, with a p of 0.04. According to your comments, we added these additional data also in the results section (line 184-187).

Results in the abstract are in general not very clear. I would present ASSOCIATIONs between the different predictors and mortality, one by one, age, gender, drugs, time of drug administration, age category... with ORs of the most relevant ones.

We rephrase the sentence in the abstract: Multivariate analysis showed that an older age (OR: 2.54) and dyspnea as presenting symptom (OR: 2.01) were associated with higher mortality, while cough as first symptom (OR: 0.53) and a timely drug administration (OR: 0.44) with lower mortality.

2- Itroduction

First sentence is too long and difficult to understand

We rephrase the sentence: During the severe acute respiratory syndrome coronavirus 2 (SARS-CoV-2) pandemic, the causal agent of coronavirus disease 2019 (COVID-19), several drugs were investigated in randomized trials worldwide including antiviral agents and antimalarial medications (https://www.fda.gov/drugs/emergency-preparedness-drugs/coronavirus-covid-19-drugs. Accessed 19 September 2020). However,

I don't understand the last lie of the intro. You look at the association between drugs and development of syptoms? Please clarify

We rephrase the sentence: Association between symptom type at disease onset, whether cough, dyspnea, fever, gastroenteric symptoms, and/or anosmia and ageusia and mortality were also tested as secondary endpoints.

3- Material and methods

Please avoid the ream retrospective. Clinicians collect the data prospectively. Refer to you study as observational.

We removed the term retrospective.

4- Results

Why do you take data only of 143 patients? I don't understand. Please clarify.

We focused our analyses on patients aged > 65 years-old for whom date of symptom onset was available.

These 143 patients were all admitted in the hospital wards? It is unclear to me what happened with those who were discharged home directly from the ED. Are they included in the study?

Our analysis focused on patients who were severe enough to be hospitalized and followed-up in hospital.

I can't see in table 1 the proportions of those who had HCQ, alone or in combination, in each age category.

It is specified in table 1: in the total group, 97 were treated with LPV/r and 46 with DVR/c (67.8% of the total). After stratification, in the 65-80 years group 68 were treated with LPV/r and 30 with DVR/c (69.4% of the total), in the >80 years group 29 were treated with LPV/r and 16 with DVR/c (64.4% of the total).

You don't mention the HCQ at all in your results. Was it not one of your drugs of interest?

HCQ was given to all patients, so we cannot draw any conclusion on its impact on mortality.

5- Discussion

Avoid the term retrospective.

We removed the term retrospective.

You have to mention that you have observed outcomes on patients receiving a combination of drugs Maybe the effects you observe come only from one of them.

We made several changes to the discussion, as you suggested. According to your comments, in line 247-248 we added the sentence: We are not aware of any study testing the efficacy of an early administration of HCQ and anti-retroviral agents, even if out of a controlled clinical trial setting. In line 291-292: Since results shown are due to the use of a combination of drugs, it will be important to understand whether the efficacy come from the early use of either drugs or only one of them.

Please include the large systematic reviews of observational studies and clinical trials that are already available on this topic, in your bibliography. You cite very few studies, selected for unclear reasons.

According to your comments, we largely modified the discussion and the references by adding and mentioning several articles on clinical trials and observational studies performed on the use of anti-viral drugs and hydroxychloroquine in COVID-19.

Reviewer 2 Report

In the reviewed manuscript by Desai et al., the effect of the administration time of antiviral treatment on mortality in elderly COVID-19 patients is evaluated. In addition, predictors in the COVID-19 hospitalized patients from February 21st to April 14th 2020 are presented through a observational study in Emergency Department of Humanitas Clinical and Research Centre, Milan, Italy.

This paper is well written, correctly structured and with a suitable research concept. The study limitations are addressed, especially the limited sample size.

Definitely, this paper is of relevance to readers of the journal, and the general findings are of importance to improve knowledge of clinical behavior of the COVID-19.

However, some suggested changes that may improve the paper are included in the comments given below.

  • The acronym COVID-19 appears type in several diverse ways throughout the report, for example, COVID-19 (line 52), Covid-19 (line 93) or COVID19 (line 187). That is why I think it appropriate unifying the way to write it.
  • Line 94: The link is not work.
  • It would be appropriate that the categorical variables represented in terms of frequency distribution are accompanied by their corresponding confidence intervals.
  • Line 148: It would be appropriate to replace "fraction" by "percentage".
  • Lines 150-151: Please order COVID-19 symptoms in decreasing order.
  • Lines 153-154: The data included in these lines does not appear in table 1.
  • Line 157: In this context, referring about trend is not the most appropriate; please consider replacing "trend" by "incidence" or other similar term.
  • Many of the data included in Table 1 are repeated in the text of the "Results" section, which is redundant.
  • Line 169: The "time to ED admission" is cited as a variable associated with mortality at univariate analysis, however this variable does not appear in Table 2, nor is it defined in the "Material and Methods" section. It would be necessary to clarify this aspect.
  • Line 178: “Interestingly enough” could be removed. The "Results" section should not contain opinions of the authors, just describe the findings aseptically.
  • Figure 1: the figure is not fully translated into English. There are undefined abbreviations, such as "Cum", or "Age_cl". Please describe them in the legend or modify the titles.
  • Figure 2: The abbreviation "Cum" is not defined. Should be included in the legend or modify the title of the y-axis.
  • Figure 3: Figure 3 is not comprehendible at the current format. Among other things, it would be necessary to correctly define the title of the y-axis, eliminate the title at the top of the figure ("Bar chart") and eliminate the underscores from the title of the x-axis.
  • The explanatory titles of the all figures are too long and repetitive with the legends content, so it is recommended to simplify the titles.
  • The last paragraph of the "Conclusions" section is not based on your findings. Please remove it or move it to the "Discussion" section.

Author Response

In the reviewed manuscript by Desai et al., the effect of the administration time of antiviral treatment on mortality in elderly COVID-19 patients is evaluated. In addition, predictors in the COVID-19 hospitalized patients from February 21st to April 14th 2020 are presented through a observational study in Emergency Department of Humanitas Clinical and Research Centre, Milan, Italy.

This paper is well written, correctly structured and with a suitable research concept. The study limitations are addressed, especially the limited sample size.

Definitely, this paper is of relevance to readers of the journal, and the general findings are of importance to improve knowledge of clinical behavior of the COVID-19.

However, some suggested changes that may improve the paper are included in the comments given below.

The acronym COVID-19 appears type in several diverse ways throughout the report, for example, COVID-19 (line 52), Covid-19 (line 93) or COVID19 (line 187). That is why I think it appropriate unifying the way to write it.

We made changes accordingly

Line 94: The link is not work.

We made changes accordingly

It would be appropriate that the categorical variables represented in terms of frequency distribution are accompanied by their corresponding confidence intervals.

We added in table 1 the OR and 95% CI of each variable, as asked. We submitted a new version of table 1, named table 1_revised.

Line 148: It would be appropriate to replace "fraction" by "percentage".

We made changes accordingly

Lines 150-151: Please order COVID-19 symptoms in decreasing order.

We made changes accordingly

Lines 153-154: The data included in these lines does not appear in table 1.

We thought as redundant to add this information in the table, since it is already described in the text.

Line 157: In this context, referring about trend is not the most appropriate; please consider replacing "trend" by "incidence" or other similar term.

We rephrase the sentence: a lower incidence of dyspnea complaint

Many of the data included in Table 1 are repeated in the text of the "Results" section, which is redundant.

We tried to reduce data shown in results as you asked.

Line 169: The "time to ED admission" is cited as a variable associated with mortality at univariate analysis, however this variable does not appear in Table 2, nor is it defined in the "Material and Methods" section. It would be necessary to clarify this aspect.

We meant to write “time to drug start”, we modified the text accordingly

Line 178: “Interestingly enough” could be removed. The "Results" section should not contain opinions of the authors, just describe the findings aseptically.

We made changes accordingly

Figure 1: the figure is not fully translated into English. There are undefined abbreviations, such as "Cum", or "Age_cl". Please describe them in the legend or modify the titles.

We modified figure 1 accordingly

Figure 2: The abbreviation "Cum" is not defined. Should be included in the legend or modify the title of the y-axis.

We modified the legend: Cum survival states for cumulative survival.

Figure 3: Figure 3 is not comprehendible at the current format. Among other things, it would be necessary to correctly define the title of the y-axis, eliminate the title at the top of the figure ("Bar chart") and eliminate the underscores from the title of the x-axis.

We modified figure 3 accordingly

The explanatory titles of the all figures are too long and repetitive with the legends content, so it is recommended to simplify the titles.

We simplified all legends content as suggested

The last paragraph of the "Conclusions" section is not based on your findings. Please remove it or move it to the "Discussion" section.

We moved the sentence from the discussion to the conclusions.

Round 2

Reviewer 1 Report

I think the study is publishable

Author Response

Dear Editor-in-chief,

We are re-submitting a revised version of the manuscript entitled “The use of antiviral agents against SARS-CoV-2: ineffective or time and age dependent result? A retrospective, observational study among COVID-19 older adults”” to the Journal of Clinical Medicine.

The manuscript was written according to your editorial requirements mentioned in the journal website.

Kindly consider and acknowledge our submission.

Filippo Martinelli Boneschi

Address: Department of Pathophysiology and Transplantation (DEPT), University of Milan, Milan, & Neurology Unit and MS Centre, Fondazione IRCCS Ca' Granda Ospedale Maggiore Policlinico, Via Francesco Sforza 35 20122 Milan, Italy. Tel: +39-3407766371; e-mail: [email protected]
